# Impact of Early Weight Catch-Up on 6-Year Neurodevelopment and Overweight/Obesity in Children Born Small-for-Gestational-Age

**DOI:** 10.3390/children13010069

**Published:** 2025-12-31

**Authors:** Hyun Ah Woo, Seon Young Kim, Eun Hye Lee, Hae Woon Jung, Eunkyo Ha, Boeun Han, Man Yong Han, Ju Hee Kim

**Affiliations:** 1Department of Pediatrics, Kyung Hee University Medical Center, Seoul 02447, Republic of Korea; 28126@khmc.or.kr (H.A.W.); 14764@khmc.or.kr (S.Y.K.); leeeh80@khmc.or.kr (E.H.L.); woonieya@gmail.com (H.W.J.); 2Department of Pediatrics, Hallym University Kangnam Sacred Heart Hospital, Seoul 07441, Republic of Korea; 3Department of Pediatrics, Bundang CHA Medical Center, CHA University School of Medicine, Seongnam 13488, Republic of Korea; a240141@chamc.co.kr; 4Multi-Omics Research Center, CHA Future Medicine Research Institute, Seongnam 13496, Republic of Korea

**Keywords:** small for gestational age, development, obesity, population-based study, children

## Abstract

**Highlights:**

**What are the main findings?**
Among children born small-for-gestational-age (SGA; <10th percentile), those with weight-for-age z-scores (WAZ) < 10th percentile at age 2 had the highest prevalence of suboptimal neurodevelopment at age 6 and a significantly increased risk. Children with WAZ ≥ 85th percentile at age 2 had the second highest prevalence; however, their risk increase was not statistically significant.Among children born SGA, those with WAZ ≥ 85th percentile at age 2 exhibited the highest prevalence of obesity at age 6 and a significantly increased risk.

**What is the implication of the main finding?**
Moderate catch-up growth by age 2 is essential for optimizing neurodevelopmental outcomes while minimizing obesity risk in children born SGA.Careful monitoring and tailored growth interventions are necessary to balance developmental and metabolic health in children born SGA.

**Abstract:**

Background/Objectives: Children born small-for-gestational-age (SGA) have varying growth patterns and developmental risks. In this study, we aimed to examine the relationship between weight-for-age z-scores (WAZ) at 2 years and neurodevelopmental and obesity outcomes at 6 years in children with SGA. Methods: We conducted a population-based cohort study using the National Health Insurance Service database of South Korea (N = 39,809). WAZ at 2 years was used to categorize children into four groups: G1 (WAZ < −1.28 [10th percentile], n = 9416), G2 (−1.28 ≤ WAZ < 0 [50th percentile], n = 20,322), G3 (0 ≤ WAZ < 1.04 [85th percentile], n = 8280), and G4 (1.04 ≤ WAZ, n = 1791). Neurodevelopment was assessed using the Korean Developmental Screening Test (K-DST). Overweight and obesity were defined using a body mass index-for-age z-score greater than the 85th and 95th percentiles, respectively, at 6 years of age. Adjusted odds ratios (aORs) and prevalence rates were estimated using Poisson and logistic regression models. Group-based comparisons were interpreted as exploratory analyses. Results: The prevalence of suboptimal neurodevelopment was highest in G1 (5.03%), followed by G4 (3.75%) at 6 years. A significantly increased risk of suboptimal K-DST scores was observed in G1 (aOR: 1.544; 95% confidence interval [CI]: 1.253–1.902), whereas a nonsignificant increase was found in G4 (aOR: 1.447; 95% CI: 0.938–2.234). At age 6, the prevalence of obesity was highest in G4 (19.60%), followed by G3 (7.11%), G2 (1.81%), and G1 (0.64%). The G4 group had the highest risk of overweight (aOR: 9.94) and obesity (aOR: 14.29) at 6 years. Conclusions: Weight status at age 2 in children with SGA was significantly associated with neurodevelopmental and obesity risks at age 6. These findings highlight the need for early weight monitoring and interventions to optimize long-term health in children with SGA.

## 1. Introduction

Infants born small-for-gestational-age (SGA), defined as having a birth weight below the 10th percentile for gestational age, face increased risks of various health and developmental challenges throughout life [1]. Among the 2.4 million global neonatal deaths in 2020, over half (55.3%) were attributable to small vulnerable newborns, with 73.4% occurring in preterm infants and the remainder in term infants with SGA [2]. The pathway to healthy newborn outcomes begins preconceptionally and continues in utero through optimal placental function, balanced nutrients, and adequate growth factors. These factors can be modulated by improving maternal nutrition, preventing infections, reducing stress, and ensuring healthy environments to mitigate risks in small vulnerable newborns [3]. Approximately 85–90% of children with SGA achieve catch-up growth during early childhood; however, the patterns and long-term consequences of this growth vary significantly depending on multiple factors [4,5]. Recent guidelines highlight the complex interplay among early weight status, developmental outcomes, and long-term health risks in children with SGA [6]. This concept is particularly relevant for infants with SGA, who may experience metabolic adaptations that influence postnatal growth trajectories and subsequent health risks [6]. Previous studies [7,8] have identified growth status at 2 years of age as a critical predictor because of its association with later obesity risk and neurodevelopmental outcomes. 

Neurodevelopmental impairments, such as lower cognitive performance, attention deficits, and an elevated risk of neuropsychiatric disorders, have been observed in both full-term [9] and preterm [10] children born SGA, particularly when they fail to achieve adequate catch-up growth. In contrast, rapid catch-up growth during infancy is known to increase the risk of metabolic disorders, including type 2 diabetes mellitus, hypertension, and cardiovascular disease [11]; elevate body mass index (BMI) during peripubertal development; and accelerate puberty timing, as evidenced by earlier menarche and peak height velocity. However, research that simultaneously evaluates the impact of catch-up growth on both neurodevelopment and obesity risk in children with SGA, particularly stratified by weight status, remains scarce. Weight-for-age before age 2 is commonly used as an indicator of nutritional status, we aimed to provide guidance on the optimal degree of weight catch-up growth at age 2.

In this study, we aimed to address this knowledge gap by examining the relationship between weight status at age 2 and neurodevelopmental and obesity outcomes at age 6 in a large longitudinal cohort of children born SGA. Furthermore, we aimed to examine how weight and developmental trajectories change from 2–6 years based on weight gain at age 2 within this population. Overall, we aimed to provide insights into optimal postnatal growth patterns that balance metabolic health and neurodevelopmental outcomes in children with SGA.

## 2. Methods

### 2.1. Study Design and Setting

In this study, we used data from the National Health Insurance Service (NHIS) database, a single-payer system covering the entire South Korean population. The NHIS database has prospectively collected data since 2008 and offers comprehensive information on healthcare utilization, including diagnoses, medication codes, procedure codes, and details of healthcare facilities visited. Its extensive coverage and detailed records make it a valuable resource for epidemiological studies, enabling large-scale population-based research with high generalizability. The National Health Screening Program for Infants and Children (NHSPIC) is a population-based surveillance initiative that provides seven rounds of screening services to all health insurance subscribers aged 4–72 months. Screenings occur at 4–6 months, 9–12 months, 18–24 months, 30–36 months, 42–48 months, 54–60 months, and 66–72 months. The NHSPIC includes physical measurements, health-related questionnaires, and developmental evaluations using the Korean Developmental Screening Test (K-DST).

The data used in this study were anonymized and de-identified before release; thus, the requirement for informed consent was waived. The study protocol was approved by the Institutional Review Board of the Korea National Institute for Bioethics Policy (P01-201603-21-005).

### 2.2. Data Sources

Demographic characteristics and healthcare utilization data—including type of visit and disease information according to the International Classification of Diseases, 10th revision (ICD-10) codes—were obtained from the NHIS database. Patients were classified by residential area of birth (Seoul; metropolitan areas [Busan, Daegu, Incheon, Gwangju, Daejeon, and Ulsan]; other cities; and rural areas) and economic status, determined by insurance copayment amounts (low: below the 25th percentile; intermediate: 25th–75th percentile; high: above the 75th percentile). Birth weight, gestational age, and breast-feeding during early infancy were recorded. 

### 2.3. Study Population

This study included 217,405 children born between 2008 and 2015 with a recorded gestational age in the NHSPIC. Children born SGA were defined as having a birth weight below the 10th percentile for gestational age [12]; 50,828 children met this criterion. Among them, 39,809 completed both the 2-year and 6-year NHSPIC examinations. To evaluate weight catch-up status at age 2, the weight-for-age z-score (WAZ) was calculated using reference values from the 2017 Korean National Growth Charts for Children and Adolescents [13], provided by the World Health Organization [14]. Children were then categorized into four groups according to their WAZ at age 2: G1 (WAZ < −1.28 [10th percentile], n = 9416), G2 (−1.28 ≤ WAZ < 0 [50th percentile], n = 20,322), G3 (0 ≤ WAZ < 1.04 [85th percentile], n = 8280), and G4 (1.04 ≤ WAZ, n = 1791) (Figure 1). The G2 group was used as the reference group.

### 2.4. Neurodevelopmental Assessment at Age 6

Developmental status at age 6 was assessed using the K-DST, a validated developmental screening tool tailored to Korean children (validation data shown in Appendix A) [15,16]. The K-DST is widely used for developmental surveillance, screening, and monitoring posttreatment changes in healthy infants. It comprises six domains: gross motor skills, fine motor skills, cognition, language, sociality, and self-care. Each domain comprises eight questions (Appendix A), scored from 0–3 points. K-DST results are interpreted across four levels based on the total score for each domain: advanced development (total score ≥ 1 standard deviation [SD] above the mean), age appropriate (−1 SD ≤ total score < 1 SD), need for follow-up (−2 SD ≤ total score < −1 SD), and recommendation for further evaluation (total score < −2 SD).

K-DST results obtained during the 7th round of the NHSPIC, conducted at age 6, were used to assess developmental status. Among the 39,809 children included in the study, K-DST results were available for 5035 children in G1, 10,416 in G2, 4111 in G3, and 882 in G4. Our primary outcome of interest was a suboptimal K-DST result, defined as a recommendation for further evaluation in any domain. A suboptimal result observed in at least one of the six domains was considered a suboptimal outcome.

### 2.5. Overweight/Obesity at Age 6

The presence of overweight or obesity at age 6 was the secondary outcome of interest. Using height and weight measured during the NHSPIC at age 6, BMI was calculated as weight (kg) divided by the square of height (m^2^). BMI-for-age z-scores were calculated using the Lambda-Mu-Sigma method described in the 2017 Korean National Growth Charts for Children and Adolescents [13]. These z-scores were then used to classify children as overweight (≥85th percentile) or obese (≥95th percentile). BMI data at age 6 were available for all children across the four WAZ groups.

### 2.6. Covariates

Covariates included sex, economic status [17], residence at birth, prematurity, birth weight, and breast-feeding during early infancy. Neonatal intensive care unit (NICU) admission history was identified using NICU admission codes, whereas diagnoses of respiratory and cardiovascular disorders specific to the perinatal period, congenital malformations, and chromosomal abnormalities were determined using ICD codes. 

### 2.7. Statistical Analysis

Baseline characteristics were summarized using descriptive statistics. Continuous variables were reported as mean ± SD or median with interquartile range, depending on their distribution. Categorical variables were presented as counts and percentages n (%). The prevalences of outcomes were summarized as n (%) across the four WAZ groups. Adjusted prevalence values were calculated using Poisson regression models, incorporating relevant covariates such as sex, socioeconomic status, gestational age, and perinatal factors to minimize confounding. Heatmaps were subsequently generated to visually represent the adjusted prevalence results.

Adjusted odds ratios (aORs) and 95% confidence intervals (CIs) for suboptimal K-DST results and overweight/obesity were estimated using multivariate logistic regression. Comparisons between groups, using G2 as the reference group, included multiple pairwise tests. Therefore, these results were interpreted in an exploratory manner, and no formal adjustment for multiple comparisons was applied. Adjustment variables included sex, birth weight group (extremely low, low, and normal), gestational age, breastfeeding during early infancy, residence at birth, socioeconomic status, diagnoses of respiratory or cardiovascular disorders specific to the perinatal period, congenital malformations or chromosomal abnormalities, and history of NICU admission. The logistic regression model included a comprehensive set of covariates to account for potential confounders and ensure robust estimates. The analysis focused on identifying significant associations and trends across WAZ groups. To estimate predicted probabilities, linear regression was used for K-DST abnormalities, and polynomial regression was applied for obesity. WAZ at age 2 was included as a continuous predictor variable in both models. 

All statistical analyses, except heatmap generation, were conducted using SAS version 9.4 (SAS Institute, Inc., Cary, NC, USA) to ensure computational accuracy and reproducibility. Heatmap visualization was performed using Python’s Matplotlib library, version 3.6.3 (Python Software Foundation, Wilmington, DE, USA).

## 3. Results

### 3.1. Characteristics of Study Population

Characteristics of the study population are presented in Table 1. A total of 39,809 participants were enrolled, with 9416, 20,322, 8280, and 1791 in the G1, G2, G3, and G4 groups, respectively. The proportions of boys and girls were comparable across all groups. No significant differences in economic status were observed among the G1, G2, and G3 groups. However, the G4 group had a higher proportion of children with low economic status (29.5% vs. 23.5–24.6%). 

Birth weight was lowest in the G1 group (1.98 kg; SD = 0.5), followed by G2 (2.17 kg; SD = 0.4), G3 (2.29 kg; SD = 0.4), and G4 (2.34 kg; SD = 0.4). The prevalence of preterm birth was highest in the G1 group, with a greater proportion of infants born before 32 weeks of gestation than in the other groups.

NICU admission rates were highest in G1 (39.4%), followed by G2 (26.7%), G3 (21.6%), and G4 (21.8%). The prevalence of respiratory and cardiovascular disorders specific to the perinatal period, congenital malformations, and chromosomal abnormalities was also highest in G1.

### 3.2. Weight Status at 2 Years and Neurodevelopment

K-DST results across different neurodevelopmental domains according to weight status at age 2 are summarized in Table 2. The proportion of children with suboptimal total K-DST scores differed significantly across groups: 319 of 5770 children in G1 (5.53%), 310 of 12,114 (2.56%) in G2, 96 of 4810 (2.00%) in G3, and 38 of 996 (3.82%) in G4. The G1 group had a significantly increased risk for suboptimal K-DST scores (aOR: 1.544; 95% CI: 1.253–1.902). The G4 group demonstrated a higher risk in the unadjusted analysis (crude OR: 1.510; 95% CI: 1.072–2.129); however, this association was attenuated and no longer statistically significant after adjustment (aOR: 1.447; 95% CI: 0.938–2.234).

Regarding individual developmental domains, the G1 group had significantly higher odds of suboptimal results in gross motor (aOR: 1.421; 95% CI: 1.010–2.001), fine motor (aOR: 1.586; 95% CI: 1.150–2.186), cognitive (aOR: 1.390; 95% CI: 1.017–1.899), language (aOR: 1.432; 95% CI: 1.053–1.947), sociality (aOR: 1.426; 95% CI: 1.020–1.993), and self-care (aOR: 1.499; 95% CI: 1.073–2.095) domains compared with the G2 group.

The G4 group exhibited a trend toward increased risk across several developmental domains. Notably, in the unadjusted analysis, G4 showed significantly higher odds in fine motor (crude OR: 1.787; 95% CI: 1.084–2.948), sociality (crude OR: 1.729; 95% CI: 1.050–2.849), and self-care (crude OR: 1.731; 95% CI: 1.020–2.938). However, these associations were attenuated and became non-significant after adjustment, with aORs of 1.517 (95% CI: 0.781–2.948) for fine motor, 1.709 (95% CI: 0.902–3.239) for sociality, and 1.408 (95% CI: 0.876–3.330) for self-care. In contrast, the G3 group had lower odds of suboptimal results across all domains than the G2.

Figure 2A presents the adjusted prevalence of suboptimal total K-DST scores from ages 3–6. Across all ages, the G1 group consistently showed the highest adjusted prevalence of suboptimal results, whereas the G4 group exhibited slightly higher adjusted prevalence compared with G2 and G3.

Figure 3 and Appendix A shows the predicted probability of suboptimal K-DST outcomes at age 6, demonstrating a U-shaped relationship with WAZ at age 2. Specifically, the probability increased sharply as WAZ decreased below 0, reached a minimum at approximately 0, and then increased slightly as WAZ increased above 0.

### 3.3. Weight Status at 2 Years and Overweight/Obesity at 6 Years

The prevalence of overweight at age 6 was highest in the G4 group (36.74%), followed by G3 (17.67%), G2 (5.68%), and G1 (2.02%) (Table 3). Compared with G2, the G4 group had a significantly increased risk of being overweight at age 6 (aOR: 9.938; 95% CI: 8.683–11.375), followed by G3 (aOR: 3.471; 95% CI: 3.156–3.817). In contrast, G1 had a significantly lower risk of being overweight at age 6 (aOR: 0.310; 95% CI: 0.257–0.374).

The prevalence of obesity at age 6 was highest in the G4 group (19.60%), followed by G3 (7.11%), G2 (1.81%), and G1 (0.64%). Compared with G2, the G4 group had the highest risk of obesity at age 6 (aOR: 14.287; 95% CI: 11.904–17.147), followed by G3 (aOR: 4.069; 95% CI: 3.489–4.744). The G1 group had a significantly lower risk of obesity at age 6 (aOR: 0.275; 95% CI: 0.194–0.390).

The adjusted prevalence of obesity from ages 3 to 6 is shown in Figure 2B. Obesity prevalence was significantly higher in G4, followed by G3, G2, and G1 in descending order.

Figure 3 shows the predicted probability of obesity at age 6 according to WAZ at age 2. The predicted probability increased progressively with WAZ, rising sharply when WAZ exceeded approximately 1.

## 4. Discussion

In this study, we analyzed data from a longitudinal nationwide cohort to assess the effect of early weight status on developmental outcomes and obesity risk in children born SGA. Infants with WAZ below the 10th percentile (G1) at age 2 had an increased risk of suboptimal development in all domains, whereas those with WAZ greater than or equal to the 85th percentile (G4) had an increased risk of overweight and obesity at age 6. Across 3–6 years, G4 demonstrated the second greatest prevalence of suboptimal neurodevelopment after G1. More than a quarter of children in G4 at age 2 remained overweight at age 6. These findings emphasize the importance of adequate, but moderate, catch-up growth in infants with SGA.

One of the most notable findings was the U-shaped relationship between WAZ at age 2 and neurodevelopmental outcomes at age 6. Children in G1 had the highest prevalence and odds of suboptimal K-DST outcomes across all developmental domains, consistent with previous research showing that inadequate catch-up growth in full-term infants with SGA and preterm infants increases the risk of cognitive and developmental delays [9,18,19]. Interestingly, children in G4 showed a trend toward increased risk of suboptimal neurodevelopment, although this association was not statistically significant. The G3 group (0 ≤ WAZ < 1.04 [85th percentile]) had the lowest prevalence and risk of suboptimal neurodevelopmental outcomes, suggesting that moderate catch-up growth may be optimal.

Our results revealed a strong association between early weight status and obesity risk in children born SGA. Children in G4 at age 2 had a significantly increased risk of both overweight and obesity at age 6, with aORs of 9.938 and 14.287, respectively. This finding aligns with the concept of early adiposity rebound, in which rapid early weight gain in infants with SGA can lead to increased fat deposition and long-term metabolic disease risk [11].Conversely, children in G1 at age 2 had a significantly lower risk of obesity at age 6. This may be beneficial for obesity prevention; however, it can compromise neurodevelopment and pose other health risks, as observed in preterm infants born SGA [20]. The G3 group showed an intermediate risk of overweight/obesity at age 6, further supporting the notion that moderate catch-up growth may optimally balance developmental and metabolic outcomes.

Our findings have important implications for the management and follow-up of children born SGA, emphasizing the importance of monitoring weight gain patterns during the first 2 years of life. Furthermore, we found that the difference in WAZ at age 2 has a more pronounced impact on the prevalence of overweight (2.02% vs. 36.74%) and obesity (0.64% vs. 19.60%) than on suboptimal neurodevelopmental screening results (2.00% vs. 5.53%) at age 6. Unlike previous studies comparing SGA and appropriate-for-gestational-age infants [10,21,22], our study focused on identifying optimal catch-up growth patterns within the SGA population. We used WAZ, a pragmatic and widely used growth indicator, and defined SGA and catch-up growth at the 10th percentile, applying conservative criteria to avoid missing children at increased health risk, particularly in developing countries. These results highlight the need for a nuanced approach to growth monitoring that balances neurodevelopmental outcomes and obesity risk. We recommend targeting a weight above the 10th percentile by age 2 while avoiding excessive weight gain. Tailored interventions for underweight children to support growth and neurodevelopment, along with early obesity prevention for those with rapid weight gain, are essential. This strategy optimizes neurodevelopmental outcomes while mitigating metabolic risks.

A major strength of our study is its large, population-based design and the use of standardized neurodevelopmental and anthropometric assessments, which enhance the robustness and generalizability of our findings. However, some limitations should be noted. A more stratified analysis of infants born SGA below the 3rd percentile could strengthen result interpretation and clarify whether associations are driven by this higher-risk subgroup; however, a previous Korean study found no significant differences in postnatal catch-up growth up to 24 months between the <3rd and 3rd–10th percentile groups [23]. More detailed subcategorization of weight groups at age 2, such as including a WAZ ≥ 95th percentile subgroup, might have influenced the neurodevelopmental outcomes observed at age 6. Preterm infants born SGA may continue catch-up growth after 3–4 years [5,24]; however, we used WAZ at age 2 as our primary criterion. These decisions were mitigated by adjusting for preterm status, NICU admission, and other comorbidities, and by performing outcome assessment at age 6, when most catch-up growth is completed, even for preterm infants [25]. 

Despite these adjustments, the study population likely represents a mixed group of infants born SGA, including those whose small size resulted from environmental factors and those with underlying genetic variation, as genetically small infants were not excluded. Furthermore, the omission of birth length and head circumference measurements, which reflect distinct intrauterine growth adaptation patterns and are linked to divergent neurodevelopmental and metabolic risk profiles, represents a limitation. Early infant breastfeeding was considered; however, detailed nutritional data on feeding practices, such as duration of exclusive breastfeeding, formula use, and complementary food quality, were not included. Important potential confounders—including parental BMI, parental height, maternal hypertension, and other parental characteristics that might cause intrauterine growth retardation—were not available in the NHSPIC dataset, which may have resulted in residual confounding. Growth hormone treatment was also not considered, although its impact on weight is minimal [26] and its effects on neurodevelopment in children born SGA remain controversial [27,28]. As an observational study, causality between early weight status and later outcomes cannot be established, and residual confounding factors, such as environmental chemicals, infections, or drugs, cannot be excluded. Future research incorporating detailed early life factors, including feeding practices and parental characteristics, may provide deeper insight into the mechanisms underlying these associations.

We suggest several directions for future research. Long-term follow-up of this cohort into adolescence and adulthood will help determine whether the observed associations persist and reveal additional risks. Investigating targeted interventions, such as nutritional strategies or early developmental support programs, may further optimize growth in children born SGA, particularly those from low socioeconomic status families. In addition, exploring the biological mechanisms linking early growth patterns to later outcomes—such as metabolic programming, epigenetic changes, and shifts in body composition during catch-up growth—will provide important clinical insights.

## 5. Conclusions

In conclusion, our study demonstrates that weight status at age 2 in children born SGA is significantly associated with both neurodevelopmental and obesity risks at age 6. Moderate catch-up growth (10th percentile ≤ WAZ < 50th percentile) by age 2, regardless of length/height, appears optimal. These findings emphasize the importance of careful monitoring and management of growth trajectories in children with SGA to optimize long-term health and developmental outcomes. They can also guide the development of targeted interventions to mitigate the adverse health risks associated with early growth acceleration.

## Figures and Tables

**Figure 1 children-13-00069-f001:**
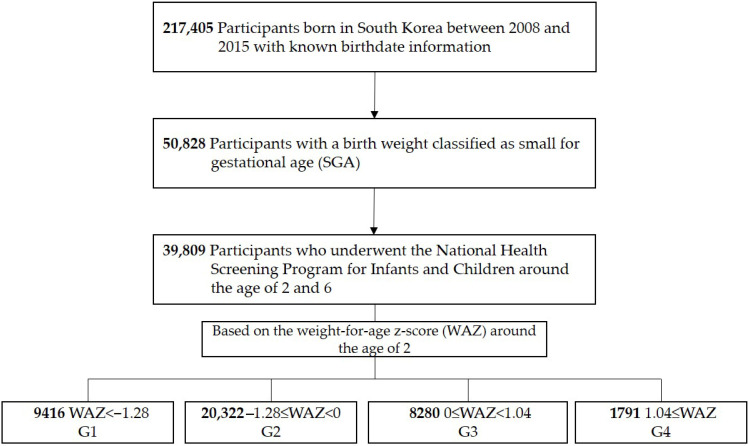
Study population.

**Figure 2 children-13-00069-f002:**
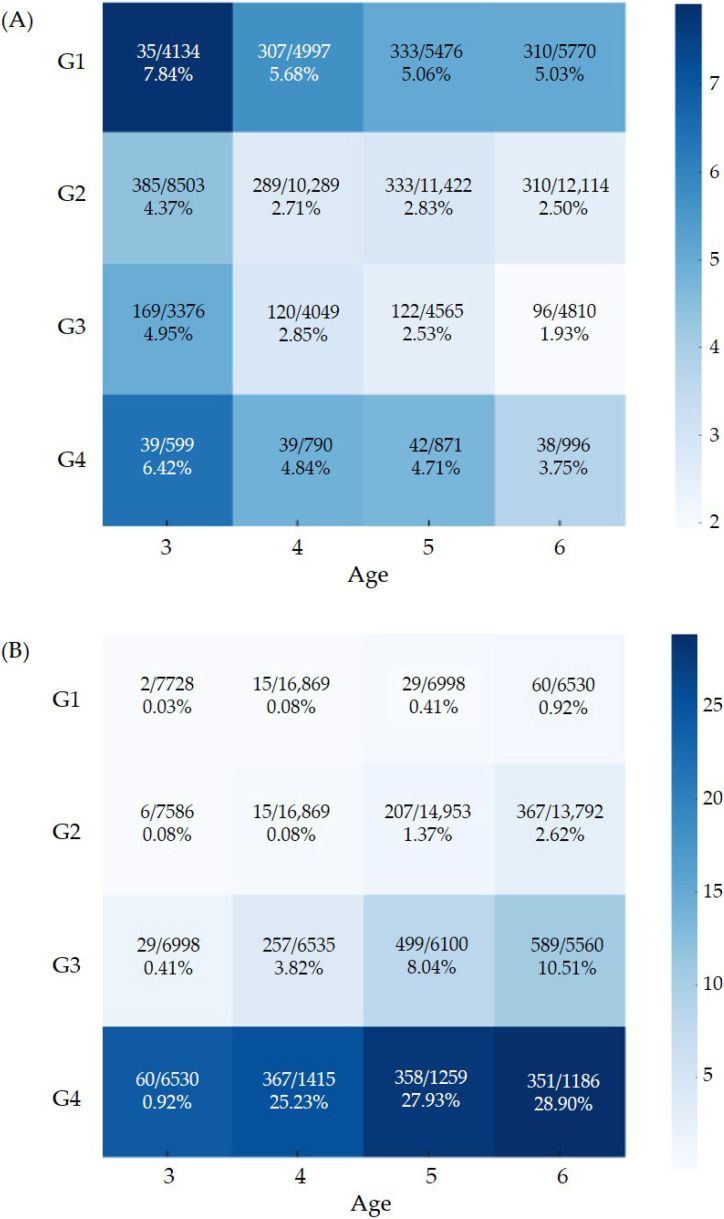
Adjusted prevalence of (**A**) suboptimal neurodevelopment screening result and (**B**) obesity. The values displayed within each heatmap cell indicate the number of events, the total number of participants, and the corresponding adjusted prevalence estimates.

**Figure 3 children-13-00069-f003:**
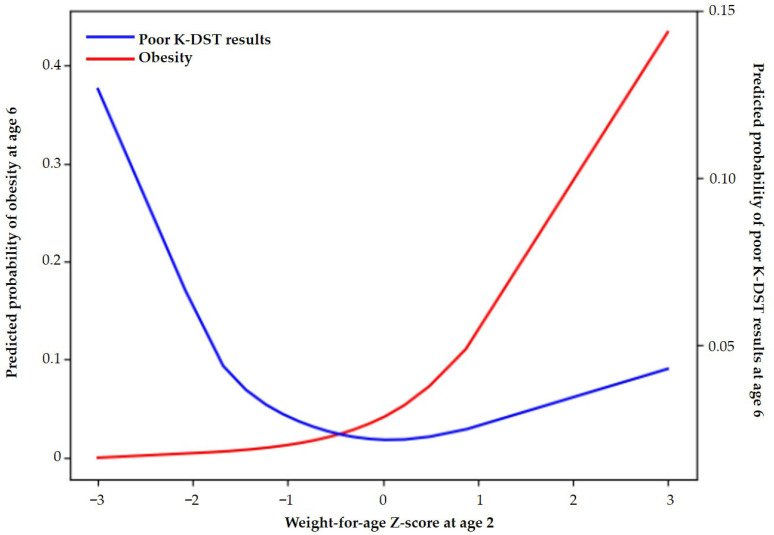
Predicted probabilities of obesity and suboptimal K-DST outcomes at age 6 according to the Z-score of weight-for-age at 2 years.

**Table 1 children-13-00069-t001:** Basic characteristics of the participants.

Characters ^a^	Total (N = 39,809)	G1 (N = 9416)	G2 (N = 20,322)	G3 (N = 8280)	G4 (N = 1791)
Sex					
Boy	19,624 (49.3)	4880 (51.8)	9837 (48.4)	3982 (48.1)	925 (51.7)
Girl	20,185 (50.7)	4536 (48.2)	10,485 (51.6)	4298 (51.9)	866 (48.4)
Birthweight, mean (SD), kg	2.16 (0.45)	1.98 (0.5)	2.17 (0.4)	2.29 (0.4)	2.34 (0.4)
Very low birth weight ^b^	2823 (7.1)	1390 (14.8)	1136 (5.6)	248 (3.0)	49 (2.7)
Low birth weight ^b^	27,481 (69.0)	6677 (70.9)	14,454 (71.1)	5288 (63.9)	1062 (59.3)
Normal birth weight ^b^	9505 (23.9)	1349 (14.3)	4732 (23.3)	2744 (33.1)	680 (38.0)
Prematurity	34,184 (85.9)	8647 (91.8)	17,456 (85.9)	6673 (80.6)	1408 (78.6)
37 weeks or more	24,671 (62.0)	5029 (53.4)	12,603 (62.0)	5756 (69.5)	1283 (71.6)
32 to 36 + 6 weeks	14,478 (36.4)	3983 (42.3)	7513 (37.0)	2477 (29.9)	505 (28.2)
31 + 6 weeks or less	278 (0.7)	146 (1.6)	105 (0.5)	25 (0.3)	2 (0.1)
Residence area at birth ^c^					
Seoul	8761 (22.0)	1826 (19.4)	4645 (22.9)	1881 (22.7)	409 (22.8)
Metropolitan	9977 (25.1)	2379 (25.3)	5129 (25.2)	2062 (24.9)	407 (22.7)
City	17,921 (45.0)	4446 (47.2)	9010 (44.3)	3659 (44.2)	806 (45.0)
Rural	2748 (6.9)	658 (7.0)	1349 (6.6)	597 (7.2)	144 (8.0)
Economic status ^d^					
Low	9624 (24.2)	2313 (24.6)	4781 (23.5)	2001 (24.2)	529 (29.5)
Intermediate	19,297 (48.5)	4605 (48.9)	9835 (48.4)	3998 (48.3)	859 (48.0)
High	9638 (24.2)	2188 (23.2)	5094 (25.1)	2008 (24.3)	348 (19.4)
Breastmilk feeding during early infancy	6718 (16.9)	1840 (19.5)	3579 (17.6)	1130 (13.7)	169 (9.44)
History of NICU admission	11,309 (28.4)	3713 (39.4)	5415 (26.7)	1790 (21.6)	391 (21.8)
Change in weight-for-age z-score from birth to around 2 years of age, median (IQR)	1.49 (0.78, 2.30)	0.34 (−0.13, 0.99)	1.36 (0.94, 1.95)	2.31 (1.93, 2.89)	3.32 (2.89, 4.05)
Respiratory and cardiovascular disorders specific to the perinatal period	8675 (32.4)	3825 (40.6)	4236 (20.8)	1625 (19.6)	319 (17.8)
Congenital malformation	12,917 (32.4)	3825 (40.6)	6243 (30.7)	2342 (28.3)	507 (28.3)
Chromosomal abnormalities	444 (1.1)	265 (2.8)	136 (0.7)	33 (0.4)	10 (0.6)

Abbreviations: N, number; SD, standard deviation; NICU, neonatal intensive care unit; IQR, interquartile range. Children were classified into four groups based on weight for age z-score at age 2: G1 (<10th percentile), G2 (10th–<50th percentile), G3 (50th–<85th percentile), and G4 (≥85th percentile). ^a^ Results are reported as N (%) unless otherwise indicated. ^b^ Very low birth weight was defined as a birth weight of less than 1500 g, low birth weight as a birth weight between 1500 g and 2499 g, and normal birth weight as a birth weight of 2500 g or more. ^c^ Metropolitan areas were defined as six metropolitan cities (Busan, Incheon, Gwangju, Daejeon, Daegu, and Ulsan), urban areas as cities, and rural areas as non-city areas. Missing data = 402. ^d^ Economic status was categorized into the quintile of insurance premiums at birth. Missing data = 1250.

**Table 2 children-13-00069-t002:** Risk of poor neurodevelopmental screening results assessed using the K-DST at 6 years.

K-DST Domain	Group ^a^	Total Number	Event, N (%)	Crude OR (95% CI)	Adjusted OR (95% CI) ^b^
Total score	G1	5770	319 (5.53)	2.229 (1.900–2.614)	1.544 (1.253–1.902)
	G2	12,114	310 (2.56)	Ref	Ref
	G3	4810	96 (2.00)	0.775 (0.615–0.977)	0.858 (0.650–1.131)
	G4	996	38 (3.82)	1.510 (1.072–2.129)	1.447 (0.938–2.234)
Gross motor	G1	5035	130 (2.58)	2.417 (1.874–3.117)	1.421 (1.010–2.001)
	G2	10,416	113 (1.08)	Ref	Ref
	G3	4111	37 (0.90)	0.828 (0.570–1.202)	0.853 (0.533–1.366)
	G4	882	16 (1.81)	1.685 (0.993–2.857)	1.770 (0.907–3.454)
Fine motor	G1	5035	147 (2.92)	2.581 (2.023–3.293)	1.586 (1.150–2.186)
	G2	10,416	120 (1.15)	Ref	Ref
	G3	4111	41 (1.00)	0.865 (0.605–1.235)	0.936 (0.612–1.433)
	G4	882	18 (2.04)	1.787 (1.084–2.948)	1.517 (0.781–2.948)
Cognition	G1	5035	143 (2.84)	2.13 (1.684–2.965)	1.390 (1.017–1.899)
	G2	10,416	141 (1.35)	Ref	Ref
	G3	4111	42 (1.02)	0.752 (0.532–1.064)	0.795 (0.523–1.209)
	G4	882	18 (2.04)	1.518 (0.925–2.492)	1.415 (0.752–2.664)
Language	G1	5035	150 (2.98)	2.116 (1.683–2.661)	1.432 (1.053–1.947)
	G2	10,416	149 (1.43)	Ref	Ref
	G3	4111	41 (1.00)	0.694 (0.490–0.983)	0.792 (0.519–1.210)
	G4	882	18 (2.04)	1.436 (0.876–2.353)	1.633 (0.889–3.000)
Sociality	G1	5035	125 (2.48)	2.113 (1.645–2.715)	1.426 (1.020–1.993)
	G2	10,416	124 (1.19)	Ref	Ref
	G3	4111	35 (0.85)	0.713 (0.489–1.039)	0.725 (0.451–1.166)
	G4	882	18 (2.04)	1.729 (1.05–2.849)	1.709 (0.902–3.239)
Self-care	G1	5035	133 (2.64)	2.543 (1.971–3.281)	1.499 (1.073–2.095)
	G2	10,416	110 (1.06)	Ref	Ref
	G3	4111	36 (0.88)	0.828 (0.567–1.208)	0.932 (0.590–1.459)
	G4	882	16 (1.81)	1.731 (1.020–2.938)	1.408 (0.876–3.330)

Abbreviations: K-DST, Korean-developmental screening test; N, number; OR, odds ratio; WAZ, weight for age z score; Ref, reference. ^a^ Groups were classified based on the weight-for-age z-score at age 2 as follows: G1 (WAZ < −1.28, n = 9416), G2 (−1.28 ≤ WAZ < 0, n = 20,322), G3 (0 ≤ WAZ < 1.04, n = 8280), and G4 (WAZ ≥ 1.04, n = 1791). ^b^ Adjusted for sex, birthweight group, gestational age, breast milk feeding during early infancy, residence at birth, socioeconomic status, diagnoses of respiratory or cardiovascular disorders specific to the perinatal period, congenital malformations or chromosomal abnormalities, and history of NICU admission.

**Table 3 children-13-00069-t003:** Risk of overweight and obesity at 6 years.

Overweight/Obesity at 6 Years ^a^	Group ^b^	Total Number	Event, N (%)	Crude OR (95% CI)	Adjusted OR (95% CI) ^c^
Overweight	G1	9416	190 (2.02)	0.342 (0.293–0.399)	0.310 (0.257–0.374)
	G2	20,322	1155 (5.68)	Ref	Ref
	G3	8280	1463 (17.67)	3.561 (3.281–3.866)	3.471 (3.156–3.817)
	G4	1791	658 (36.74)	9.641 (8.611–10.793)	9.938 (8.683–11.375)
Obesity	G1	9416	60 (0.64)	0.349 (0.268–0.459)	0.275 (0.194–0.390)
	G2	20,322	367 (1.81)	Ref	Ref
	G3	8280	589 (7.11)	4.164 (3.646–4.756)	4.069 (3.489–4.744)
	G4	1791	351 (19.60)	13.253 (11.341–15.488)	14.287 (11.904–17.147)

Abbreviations, N, number; OR, odds ratio; WAZ, weight for age z score; Ref, reference. ^a^ Body weight category was classified based on the body mass index for age z score at age 6: overweight (>1.04) and obesity (>1.64). ^b^ Groups were classified based on the weight-for-age z-score at age 2 as follows: G1 (WAZ < −1.28, n = 9416), G2 (−1.28 ≤ WAZ < 0, n = 20,322), G3 (0 ≤ WAZ < 1.04, n = 8280), and G4 (WAZ ≥ 1.04, n = 1791). ^c^ Adjusted for sex, birthweight group, gestational age, breast milk feeding during early infancy, residence at birth, socioeconomic status, diagnoses of respiratory or cardiovascular disorders specific to the perinatal period, congenital malformations or chromosomal abnormalities, and history of NICU admission.

## Data Availability

This study was based on the National Health Claims Database established by the NHIS of the Republic of Korea. Applications for using NHIS data are be reviewed by the Inquiry Committee of Research Support; if the application is approved, raw data is provided to the applicant for a fee. We cannot provide access to the data, analytic methods, and research materials to other researchers because of the intellectual property rights of this database that is owned by the National Health Insurance Corporation. However, investigators who wish to reproduce our results or replicate the procedure can be used in the database, which is open for research purposes (https://nhiss.nhis.or.kr/, accessed on 12 July 2024).

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
