# Peer review of "Impact of Early Weight Catch-Up on 6-Year Neurodevelopment and Overweight/Obesity in Children Born Small-for-Gestational-Age"

_children, 2025, doi:10.3390/children13010069_

Round 1
Reviewer 1 Report
Comments and Suggestions for Authors
This is an interesting and well-conducted population-based study examining the association between early weight status and later developmental and obesity outcomes in small-for-gestational-age (SGA) children. The topic is important and relevant for both pediatric and public health practice. The manuscript is generally clear and well structured.
Major comments
-
Please clarify the flow of participants across the study stages (initial SGA cohort, those assessed at 2 years, and those with available data at 6 years). A short flow diagram or additional description would improve transparency.
-
The rationale for defining the WAZ categories, particularly the upper cut-off of 1.04, could be briefly explained.
-
It would be helpful to specify the main covariates included in the adjusted models and clarify whether odds ratios or risk ratios were estimated.
-
Please acknowledge key potential confounders not available in the dataset (e.g., parental BMI, feeding patterns, maternal conditions) and how these limitations might affect interpretation.
-
Figures are clear, but adding confidence intervals or sample sizes where possible would make them more informative.
Minor comments
-
Ensure consistent use and definition of abbreviations (WAZ, K-DST, NHSPIC).
-
A light language edit for grammar and style would further improve readability.
-
Check formatting and table alignment for clarity.
Overall, this is a valuable contribution that provides insight into the optimal range of catch-up growth for SGA children. With these clarifications and minor revisions, the manuscript will be suitable for publication.
Author Response
Response to Reviewer 1
This is an interesting and well-conducted population-based study examining the association between early weight status and later developmental and obesity outcomes in small-for-gestational-age (SGA) children. The topic is important and relevant for both pediatric and public health practice. The manuscript is generally clear and well structured.
Comment 1: Please clarify the flow of participants across the study stages (initial SGA cohort, those assessed at 2 years, and those with available data at 6 years). A short flow diagram or additional description would improve transparency.
Reply 1: Thank you for this helpful comment. We agree that clearly describing the flow of participants across the study stages is essential for transparency. In response, we have added a detailed description of the participant flow in the Methods section, specifying the number of children in the initial SGA cohort, those who completed the NHSPIC at age 2, and those with available outcome data at age 6. As clarified, 39,809 children completed both the 2-year and 6-year NHSPIC examinations and were included in the final analysis.
[Revised version of Figure 1]
Among them, BMI data were available for all participants across the four WAZ groups, while K-DST developmental outcomes were available for 5,035 children in G1, 10,416 in G2, 4,111 in G3, and 882 in G4.
To further enhance transparency, we have also included a flow diagram (Figure 1) summarizing the inclusion process from the initial 217,405 children to the final analytic sample.
Line 119-129
This study included 217,405 children born between 2008 and 2015 with a recorded gestational age in the NHSPIC. SGA was defined as a birth weight below the 10th percentile for gestational age (12), and 50,828 children met this criterion. Among them, 39,809 completed both 2-year and 6-year NHSPIC examinations. To evaluate weight catch-up status at 2 years of age, the weight-for-age z-score (WAZ) at 2 years was calculated using reference values from the 2017 Korean National Growth Charts for Children and Adolescents (13), provided by the World Health Organization (14). Children were then categorized into four groups according their WAZ at 2 years: G1 (WAZ <−1.28 [10th percentile], N = 9,416), G2 (−1.28≤ WAZ <0 [50th percentile], N = 20,322), G3 (0≤ WAZ <1.04 [85th percentile], N = 8,280), and G4 (1.04≤ WAZ, N = 1,791) (Figure 1). Children in the G2 group were used as the reference group.
Line 143-148
K-DST results obtained during the 7th round of the NHSPIC performed at 6 years of age were used to assess developmental status. Among the 39,809 children included in the study, K-DST results were available for 5,035 children in G1, 10,416 in G2, 4,111 in G3, and 882 in G4.
Line 155-156
BMI data at age 6 were available for all children across the four WAZ groups.
Thank you for your comment. We used a Weight-for-Age Z-score (WAZ) of 1.04 (85th percentile) for Group 4. According to the WHO WAZ percentile stratification, the 85th percentile was used. Although the upper Z-score cutoff is typically defined as above 2 SD, we used the 85th percentile instead because our goal is to determine the outcome of overweight/obesity based on the 6-year BMI-Z-score. We provided the percentile next to the Z-score upper cutoff in the Methods section and beyond. (https://www.who.int/tools/child-growth-standards/standards/weight-for-age) We are also aware of that since WAZ is mostly used in catch up growth or undernutrition status, the usage of WAZ as overnutrition status is a unique standard of this paper to emphasize the risk of weight over-catch up. Thank you very much for bringing this important point and we revised the manuscript as follows.
Line 80-84
However, research simultaneously analyzing the impact of catch-up growth on both neurodevelopment and obesity risk in SGA children, particularly stratified by weight status alone, is scarce. Although weight-for-age before age 2 is most commonly used as a reflection of nutritional status, our study aims to provide guidance on the optimal degree of weight catch-up growth at age 2.
Comment 3: It would be helpful to specify the main covariates included in the adjusted models and clarify whether odds ratios or risk ratios were estimated.
Reply3: Thank you for your comment. We have clarified in the Methods section that aORs were estimated using multivariate logistic regression. We also specified the covariates included in the adjusted models, which were selected based on established literature and clinical relevance. These covariates included sex, birthweight group, gestational age, breast milk feeding, residence at birth, socioeconomic status, perinatal respiratory or cardiovascular disorders, congenital malformations or chromosomal abnormalities, and history of NICU admission. For readability, all adjustment variables have also been summarized in a footnote below the relevant tables. In addition, as suggested, we have added the crude (unadjusted) odds ratios to the tables so that readers can clearly compare crude and adjusted estimates.
Line 173-178
Adjusted odds ratios (aORs) and 95% confidence intervals (CIs) for poor K-DST results and overweight/obesity were estimated using multivariate logistic regression. The adjustment variables included sex, birthweight group (extremely low birth weight, low birth weight, and normal birth weight), gestational age, breast milk feeding during early infancy, residence at birth, socioeconomic status, diagnoses of respiratory or cardiovascular disorders specific to the perinatal period, congenital malformations or chromosomal abnormalities, and history of NICU admission.
Comment 4: Please acknowledge key potential confounders not available in the dataset (e.g., parental BMI, feeding patterns, maternal conditions) and how these limitations might affect interpretation.
Reply 4: Thank you for pointing out this important issue. As the NHSPIC dataset does not include several key potential confounders—such as parental BMI, detailed infant feeding patterns, and maternal health conditions—we acknowledge that residual confounding may remain. We have added this limitation to the manuscript to clarify that these unmeasured factors may influence the interpretation of the associations observed in our study.
Line 357-360
Additionally, important potential confounders—such as parental BMI, parental height, maternal metabolic conditions, and other parental characteristics—were not available in the NHSPIC dataset, and these unmeasured factors may have resulted in residual confounding. Also, growth hormone treatment was not considered, but its impact on weight is minimal (26) and its effects on neurodevelopment in SGA children remain controversial (27, 28). As an observational study, causality between early weight status and later out-comes cannot be established, and residual confounding factors such as environmental chemicals (29), infections (30), or drugs (31) cannot be excluded. Future research incorpo-rating more detailed early life factors, including feeding practices (32, 33) and parental characteristics, may provide deeper insight into the mechanisms behind these associations.
Comment 5: Figures are clear, but adding confidence intervals or sample sizes where possible would make them more informative.
Reply 5: Thank you for this helpful suggestion. To enhance the informativeness of the figures, we have added the total number of participants and the number of events within each cell of the heatmap. We appreciate your guidance, which has contributed to improving the clarity and interpretability of our results.
Figure 2. Adjusted prevalence of (A) poor neurodevelopment screening result and (B) obesity. The values displayed within each heatmap cell indicate the number of events, the total number of participants, and the corresponding adjusted prevalence estimates.
Minor comments
: Thank you for your suggestions. We will make sure to check this again.
: Thank you for your suggestions. Although we had the manuscript professionally edited in English once before, we will carefully review the grammar again with the revised version.
Check formatting and table alignment for clarity.
: Yes, we have reviewed and corrected the formatting and table alignment for clarity.

Reviewer 2 Report
Comments and Suggestions for Authors
This study addresses a clinically meaningful and often underappreciated population. By focusing specifically on children born small for gestational age (SGA) within a large, nationwide cohort, the authors provide valuable data that reinforce the importance of early-life monitoring in a group that is frequently perceived as having achieved “catch-up” and therefore risks being overlooked in routine follow-up. The simultaneous consideration of neurodevelopmental outcomes and obesity risk represents a notable strength and offers important insights for clinicians caring for these children. I have few suggestions to the manuscript:
- Defining SGA solely on the basis of birth weight below the 10th percentile inevitably groups together infants with heterogeneous etiologies, severities of intrauterine growth restriction, and postnatal growth trajectories. Prior literature indicates that children with more severe growth restriction (birthweight SDS < −2) contribute disproportionately to adverse neurodevelopmental and metabolic outcomes. Given the large sample size of the present cohort, an additional stratified analysis focusing on infants with SDS < −2 would considerably strengthen the interpretation of the results and clarify whether observed associations are driven by this higher-risk subgroup.
- In contrast, differentiation between symmetric and asymmetric SGA phenotypes—when data on birth length and head circumference are unavailable—should be explicitly acknowledged as a limitation. These phenotypes reflect distinct intrauterine adaptations and are known to carry different neurodevelopmental and metabolic risk profiles. Failure to distinguish them may obscure important biologic differences within the cohort and should be clearly stated when interpreting the findings.
- While weight-for-age is a pragmatic and widely used growth indicator in large population-based datasets, weight alone does not adequately distinguish between linear growth and adiposity, nor does it reflect body composition. This distinction is particularly relevant given that one of the primary outcomes of the study is overweight and obesity at later childhood. Given that, the use of weight-for-height z-scores—or alternatively BMI-for-age—would represent an appropriate primary metric for early-life growth assessment than weight-for-age.
- In addition, interpretation of early weight trajectories would benefit from acknowledgment of the limited availability of detailed nutritional data. While breastfeeding during early infancy was considered, more information regarding feeding practices—such as duration of exclusive breastfeeding, formula feeding, timing and quality of complementary foods, and overall nutritional composition—was not available. Explicitly recognizing this limitation would strengthen the discussion by clarifying that observed weight trajectories may reflect a combination of nutritional, environmental, and biological influences, rather than growth patterns alone,
Overall, this study provides important and clinically relevant evidence underscoring the need for continued surveillance of children born SGA beyond infancy. Greater emphasis on the inherent heterogeneity of the SGA population, the role of early nutritional factors, and the use of height-adjusted growth indicators for early-life assessment would further enhance the interpretability and clinical relevance of the findings.
Author Response
Response to Reviewer 2
Comment: This study addresses a clinically meaningful and often underappreciated population. By focusing specifically on children born small for gestational age (SGA) within a large, nationwide cohort, the authors provide valuable data that reinforce the importance of early-life monitoring in a group that is frequently perceived as having achieved “catch-up” and therefore risks being overlooked in routine follow-up. The simultaneous consideration of neurodevelopmental outcomes and obesity risk represents a notable strength and offers important insights for clinicians caring for these children. I have few suggestions to the manuscript:
Comment 1: Defining SGA solely on the basis of birth weight below the 10th percentile inevitably groups together infants with heterogeneous etiologies, severities of intrauterine growth restriction, and postnatal growth trajectories. Prior literature indicates that children with more severe growth restriction (birthweight SDS < −2) contribute disproportionately to adverse neurodevelopmental and metabolic outcomes. Given the large sample size of the present cohort, an additional stratified analysis focusing on infants with SDS < −2 would considerably strengthen the interpretation of the results and clarify whether observed associations are driven by this higher-risk subgroup.
Reply 1: We deeply share your concern, which represents an important point we also deliberated extensively during the early phase of the study design. By defining small for gestational age (SGA) and catch-up growth at the 10th percentile, we adopted more conservative screening criteria to avoid missing children with SGA who are at increased health risk, particularly in developing countries (Line 300-302). As you correctly noted, when infants are born with a birth weight below the 3rd percentile for gestational age, heterogeneous etiologies warrant consideration; accordingly, our analysis incorporated covariates such as history of neonatal intensive care unit (NICU) admission, diagnoses of respiratory and cardiovascular disorders specific to the perinatal period, congenital malformations, and chromosomal abnormalities. Also, there is previous Korean study where two groups have no significant differences in postnatal catch-up growth up to age 24 months. (Korean J Pediatr. 2018 Mar;61(3):71-77.) Recent studies frequently adopt the INTERGROWTH-21st birthweight standard with a <10th percentile cutoff for SGA, though no consensus exists and debate persists. (Lancet. 2023 May 20;401(10389):1692-1706.) Regrettably, performing the additional analyses at this juncture would demand substantial time resources. We therefore intend to address this matter as a limitation within the Discussion section and commit to undertaking the further analyses in due course to report the findings. Should the reviewer deem these results indispensable, we respectfully invite further notification, whereupon we shall prioritize their execution.
Therefore, we included “A more stratified analysis of SGA infants born below the 3rd percentile would strengthen result interpretation and clarify whether associations are driven by this higher-risk subgroup; however, a previous Korean study found no significant differences in postnatal catch-up growth up to 24 months between the <3rd and 3rd-10th percentile groups” At Line 338-342.
Comment 2: In contrast, differentiation between symmetric and asymmetric SGA phenotypes—when data on birth length and head circumference are unavailable—should be explicitly acknowledged as a limitation. These phenotypes reflect distinct intrauterine adaptations and are known to carry different neurodevelopmental and metabolic risk profiles. Failure to distinguish them may obscure important biologic differences within the cohort and should be clearly stated when interpreting the findings.
Reply 2: We thank the reviewer for this insightful comment and fully concur with the observation. Although we mitigated this issue through covariate adjustment, birth length and head circumference may substantially influence 6-year BMI Z-scores and neurodevelopment; accordingly, we have acknowledged these as limitations in the Discussion section.
We included “Additionally, the omission of birth length and head circumference measurements, which reflect distinct intrauterine growth adaptation patterns and are linked to divergent neurodevelopmental and metabolic risk profiles, represents a limitation of this study.” At Line 352-355.
Comment 3: While weight-for-age is a pragmatic and widely used growth indicator in large population-based datasets, weight alone does not adequately distinguish between linear growth and adiposity, nor does it reflect body composition. This distinction is particularly relevant given that one of the primary outcomes of the study is overweight and obesity at later childhood. Given that, the use of weight-for-height z-scores—or alternatively BMI-for-age—would represent an appropriate primary metric for early-life growth assessment than weight-for-age.
Reply 3: We thank the reviewer for this valuable comment. The selection of appropriate postnatal growth indicators was extensively deliberated during the study design phase. Given evidence supporting the adequacy of weight-for-age Z-score (WAZ) alone for assessing infant growth until 2 years, along with its frequent use in prior studies, (J Pediatr. 2017 Mar;182:127–132.e1, Pediatrics. 2022 Aug 1;150(Suppl 1):e2022057092H.) we employed 2-year WAZ as the primary outcome measure.
However, we aware that since WAZ is mostly used in catch up growth or undernutrition status, the usage of WAZ as overnutrition status should be emphasized more. Thank you very much for bringing this important point and we included following sentences in our manuscript of introduction, discussion, conclusion as well.
Line 80-84
However, research simultaneously analyzing the impact of catch-up growth on both neurodevelopment and obesity risk in SGA children, particularly stratified by weight status alone, is scarce. Although weight-for-age before age 2 is most commonly used as a reflection of nutritional status, our study aims to provide guidance on the optimal degree of weight catch-up growth at age 2.
Line 352-355
Additionally, the omission of birth length and head circumference measurements, which reflect distinct intrauterine growth adaptation patterns and are linked to divergent neurodevelopmental and metabolic risk profiles, represents a limitation of this study.
Line 380-383
In conclusion, our study demonstrates that weight status at age 2 in children with SGA is significantly associated with both neurodevelopmental and obesity risks at age 6 and suggests moderate catch up (10th percentile ≤ WAZ < 50th percentile) by age 2 regardless of length/height.
Comment 4: In addition, interpretation of early weight trajectories would benefit from acknowledgment of the limited availability of detailed nutritional data. While breastfeeding during early infancy was considered, more information regarding feeding practices—such as duration of exclusive breastfeeding, formula feeding, timing and quality of complementary foods, and overall nutritional composition—was not available. Explicitly recognizing this limitation would strengthen the discussion by clarifying that observed weight trajectories may reflect a combination of nutritional, environmental, and biological influences, rather than growth patterns alone.
Reply 4: We thank the reviewer for this insightful comment. We agree with this suggestion and have added the following sentence to the Discussion section.
Line 355-357
Although early infant breastfeeding was considered, detailed nutritional data on feeding practices, such as duration of exclusive breastfeeding, formula use, and complementary food quality were not considered, constituting a limitation
Comment 5: Overall, this study provides important and clinically relevant evidence underscoring the need for continued surveillance of children born SGA beyond infancy. Greater emphasis on the inherent heterogeneity of the SGA population, the role of early nutritional factors, and the use of height-adjusted growth indicators for early-life assessment would further enhance the interpretability and clinical relevance of the findings.
Reply 5: We thank the reviewer for these insightful comments. We agree with this suggestion and have amended our manuscript accordingly. We hope that this change will make the content more convenient and accessible for readers. We sincerely appreciate your guidance, which has helped to improve the clarity of our results.

Reviewer 3 Report
Comments and Suggestions for Authors
The one strength of this study is its large sample size. An important concept that is missing is that those born SGA include 2 subgroups, infants who had growth restriction in utero and genetically small infants. The latter group would be expected to continue to have a shorter stature throughout their lives, that is to catch up less. Among those SGA infants with lower WAZ and those who had a lower catch up growth likely include these smaller individuals but also likely includes some with medical conditions, so it is likely a mixed group.
There are 3 big problems with this paper. 1. weight-for-age z-scores are not an appropriate measure since they do not take height into account. 2. Using −1.28 (the 10th percentile) at age 2 is not an appropriate cut point, it is not endorsed by any group at this age. It would be better to use the WHO recommended cut points: deOnis et al. 2019 PMID: 30296964. 3. >1.64 is also not a validated cut point. To assess obesity, body mass index should be used and WAZ should be ignored
The title refers to obesity but the highest WAZ category was ≥1.04 which at most should be considered overweight not obesity.
It is not appropriate to subdivide the middle group into “low-to-median” and “median-to-high” since this medicalizes appropriate sizes and could contribute to stigma, which might contribute to weight gain in children (PMID: 33079337).
I could not find how adjustment variables were decided upon nor what variables were adjusted for. This analysis does not increase understanding of how to improve outcomes. It is not clear of what importance is the risk of obesity after adjustment since variables including social, prenatal, medical variables might be important to recognize as possible contributors. The most useful way to report adjusted results is to also report the crude unadjusted results as well as the effect size for each variable in a table to show the effects of the adjustments on all the variables. Rather than focusing on adjusted odds ratios in the abstract, it would make sense to report how many children were at risk of suboptimal K-DST scores and also to report the associated variables and their strengths of association.
Author Response
Response to Reviewer 3
Comment 1: The one strength of this study is its large sample size. An important concept that is missing is that those born SGA include 2 subgroups, infants who had growth restriction in utero and genetically small infants. The latter group would be expected to continue to have a shorter stature throughout their lives, that is to catch up less. Among those SGA infants with lower WAZ and those who had a lower catch up growth likely include these smaller individuals but also likely includes some with medical conditions, so it is likely a mixed group.
Reply 1: We sincerely thank the reviewer for this excellent comment and fully agree with the point raised. While we attempted to control for these confounding factors to the greatest extent possible using the following covariates: (sex, economic status, residence at birth, prematurity, birth weight, breast-feeding during early infancy, history of neonatal intensive care unit (NICU) admission, diagnoses of respiratory and cardiovascular disorders specific to the perinatal period, congenital malformations, and chromosomal abnormalities), we acknowledge that the influence of residual confounding has not been entirely excluded. Therefore, we have added the following statement to the Discussion to explicitly address this limitation. Thank you.
Line 349-352
Although we adjusted for these covariates, the study population likely represents a mixed group of SGA infants, including those whose small size resulted solely from environmental factors and those with underlying genetic issues, as we did not exclude genetically small infants.
Comment 2: There are 3 big problems with this paper. 1. weight-for-age z-scores are not an appropriate measure since they do not take height into account. 2. Using −1.28 (the 10th percentile) at age 2 is not an appropriate cut point, it is not endorsed by any group at this age. It would be better to use the WHO recommended cut points: deOnis et al. 2019 PMID: 30296964. 3. >1.64 is also not a validated cut point. To assess obesity, body mass index should be used and WAZ should be ignored.
Reply 2: We sincerely thank the reviewer for these insightful and valuable comments. We have provided point-by-point responses to the issues raised below.
- weight-for-age z-scores are not an appropriate measure since they do not take height into account.
The World Health Organization (WHO) acknowledges Weight-for-Age Z-score (WAZ) as one of the core indicators for infants under 5 years of age along with Length/height-for-age and weight-for-length/height. (https://www.who.int/tools/child-growth-standards/standards/weight-for-age) Given evidence supporting the adequacy of weight-for-age Z-score (WAZ) alone for assessing infant growth until 2 years, along with its frequent use in prior studies, (J Pediatr. 2017 Mar;182:127–132.e1, Pediatrics. 2022 Aug 1;150(Suppl 1):e2022057092H.) we employed 2-year WAZ as the primary outcome measure.
However, we aware that since WAZ is mostly used in catch up growth or undernutrition status, the usage of WAZ as overnutrition status should be emphasized more. Thank you very much for bringing this important point and we included following sentences in our manuscript of introduction, discussion, conclusion as well.
Line 80-84
However, research simultaneously analyzing the impact of catch-up growth on both neurodevelopment and obesity risk in SGA children, particularly stratified by weight status alone, is scarce. Although weight-for-age before age 2 is most commonly used as a reflection of nutritional status, our study aims to provide guidance on the optimal degree of weight catch-up growth at age 2.
Line 352-355
Additionally, the omission of birth length and head circumference measurements, which reflect distinct intrauterine growth adaptation patterns and are linked to divergent neurodevelopmental and metabolic risk profiles, represents a limitation of this study.
Line 380-383
In conclusion, our study demonstrates that weight status at age 2 in children with SGA is significantly associated with both neurodevelopmental and obesity risks at age 6 and suggests moderate catch up (10th percentile ≤ WAZ < 50th percentile) by age 2 regardless of length/height.
- Using −1.28 (the 10thpercentile) at age 2 is not an appropriate cut point, it is not endorsed by any group at this age. It would be better to use the WHO recommended cut points: deOnis et al. 2019 PMID: 30296964.
Specifically, for analyzing catch-up growth in small, vulnerable newborns, including those born small-for-gestational-age (SGA), many previous studies have defined catch-up as achieving the 10th percentile for weight-for-age z-score (WAZ) at 2 years of age. For example, a study published in Pediatric Research (2023 Jul; 94(1):365–370) defined catch-up growth at 12, 24, and 36 months as achieving a percentile ≥10th for at least one of weight, length, or head circumference. In line with these reports, studies in the Italian Journal of Pediatrics (2021 Mar 16; 47:66) and the Korean Journal of Pediatrics (2018 Mar; 61(3):71–77) also adopted attainment of the 10th percentile for WAZ at 2 years as the criterion for catch-up growth in SGA cohorts. Therefore, the present study also stratified participants into groups based on the 10th percentile for weight-for-age Z-score (WAZ) at 2 years of age.
- >1.64 is also not a validated cut point. To assess obesity, body mass index should be used and WAZ should be ignored.
We apologize for the confusion. To determine whether children achieved catch-up growth by age 2, we used the weight-for-age z-score. For the outcomes of overweight and obesity at age 6, we assessed adiposity using the BMI-for-age z-score, where values ≥ 1.04 correspond to the 85th percentile (overweight) and values ≥ 1.64 correspond to the 95th percentile (obesity).
Comment 3. The title refers to obesity but the highest WAZ category was ≥1.04 which at most should be considered overweight not obesity.
Reply 3: As mentioned above, we defined overweight and obesity using the BMI-for-age z-scores at age 6, corresponding to the 85th and 95th percentiles, respectively. However, we agree that the previous title, which referred only to “obesity,” did not fully reflect the content of the study. Therefore, we have revised the title as follows.
Impact of Early Weight Catch-Up on Six-Year Neurodevelopment and Overweight/Obesity in Small-for-Gestational-Age Children
Comment 4: It is not appropriate to subdivide the middle group into “low-to-median” and “median-to-high” since this medicalizes appropriate sizes and could contribute to stigma, which might contribute to weight gain in children (PMID: 33079337).
Reply 4: Thank you for this valuable point. Although we initially used the median because it represented the 50th percentile cutoff for group classification, we agree that such terminology may inadvertently contribute to weight-related stigma. To avoid this, we have replaced descriptive category names with neutral, percentile-based group labels. The revised groups are defined as follows:
Low WAZ → G1 (WAZ <10th percentile)
Low-to-Median WAZ → G2 (10th ≤ WAZ <50th percentile)
Median-to-High WAZ → G3 (50th ≤ WAZ <85th percentile)
High WAZ → G4 (WAZ ≥85th percentile)
We will use these neutral group labels consistently throughout the manuscript.
Line 125-129
Children were then categorized into four groups according to their WAZ at 2 years: G1 (WAZ <−1.28 [10th percentile], N = 9,416), G2 (−1.28≤ WAZ <0 [50th percentile], N = 20,322), G3 (0≤ WAZ <1.04 [85th percentile], N = 8,280), and G4 (1.04≤ WAZ, N = 1,791) (Figure 1). Children in the G2 group were used as the reference group.
Comment 5: I could not find how adjustment variables were decided upon nor what variables were adjusted for. This analysis does not increase understanding of how to improve outcomes. It is not clear of what importance is the risk of obesity after adjustment since variables including social, prenatal, medical variables might be important to recognize as possible contributors. The most useful way to report adjusted results is to also report the crude unadjusted results as well as the effect size for each variable in a table to show the effects of the adjustments on all the variables. Rather than focusing on adjusted odds ratios in the abstract, it would make sense to report how many children were at risk of suboptimal K-DST scores and also to report the associated variables and their strengths of association.
Reply 4: Thank you for your thoughtful and constructive comment. We agree that providing transparency regarding the selection of adjustment variables and the presentation of both crude and adjusted results would improve the clarity and interpretability of our findings. We have now clarified in the Methods section that the adjustment variables were selected based on factors known to influence neurodevelopmental and physical growth in children, drawing from established literature and clinical relevance. In addition, to enhance readability and ensure that the adjusted analyses are fully transparent, we have summarized all adjustment variables in a footnote below the relevant tables. In addition, following your suggestion, we have added a table presenting the crude (unadjusted) odds ratios, adjusted odds ratios, and effect sizes for each covariate. This allows readers to clearly understand how the adjustments affect the observed associations. Third, we have revised the Results sections to include the number of children with suboptimal K-DST scores, as well as the key variables associated with these outcomes and their strengths of association. This provides a more comprehensive overview of the burden and determinants of developmental risk in children born SGA.
Methods section
Line 173-178
Adjusted odds ratios (aORs) and 95% confidence intervals (CIs) for poor K-DST results and overweight/obesity were estimated using multivariate logistic regression. The adjustment variables included sex, birthweight group (extremely low birth weight, low birth weight, and normal birth weight), gestational age, breast milk feeding during early infancy, residence at birth, socioeconomic status, diagnoses of respiratory or cardiovascular disorders specific to the perinatal period, congenital malformations or chromosomal abnormalities, and history of NICU admission.
Result section
Line 219-221
The G1 group showed a significantly increased the risk for poor K-DST scores (aOR: 1.544, 95% CI: 1.253–1.902). The G4 group also demonstrated a higher risk in the crude analysis (crude OR: 1.510, 95% CI: 1.072–2.129); however, this association was attenuated and no longer statistically significant after adjustment (aOR: 1.447, 95% CI: 0.938–2.234).
Line 236-239
The G4 group exhibited a trend toward increased risk across several developmental domains. Notably, in the crude analysis, G4 showed significantly higher odds in fine motor (crude OR: 1.787, 95% CI: 1.084–2.948), sociality (crude OR: 1.729, 95% CI: 1.050–2.849) and self-care (crude OR: 1.731, 95% CI: 1.020–2.938). However, these associations were attenuated and became non-significant after adjustment, with adjusted odds ratios of 1.517 (95% CI: 0.781–2.948) for fine motor, 1.709 (95% CI: 0.902–3.239) for sociality and 1.408 (95% CI: 0.876–3.330) for self-care. In contrast, the G3 group had lower odds of poor results across all domains compared with the G2 group.
Discussion section
Line 303-307
Without adjustment, G4 group showed poor outcomes in fine motor, sociality and self-care, and we must note that G4 group has lower economic status than other groups. This finding suggests that rapid catch-up growth leading to an over nutritious status may not necessarily confer neurodevelopmental benefits and could potentially have adverse effects on neurodevelopment.

Round 2
Reviewer 2 Report
Comments and Suggestions for Authors
Authors replied and revised the manuscript according to the suggestions